# Novel *TNXB* Variants in Two Italian Patients with Classical-Like Ehlers-Danlos Syndrome

**DOI:** 10.3390/genes10120967

**Published:** 2019-11-25

**Authors:** Lucia Micale, Vito Guarnieri, Bartolomeo Augello, Orazio Palumbo, Emanuele Agolini, Valentina Maria Sofia, Tommaso Mazza, Antonio Novelli, Massimo Carella, Marco Castori

**Affiliations:** 1Division of Medical Genetics, Fondazione IRCCS-Casa Sollievo della Sofferenza, 71043 San Giovanni Rotondo (Foggia), Italy; v.guarnieri@operapadrepio.it (V.G.); b.augello@operapadrepio.it (B.A.); o.palumbo@operapadrepio.it (O.P.); m.carella@operapadrepio.it (M.C.); m.castori@operapadrepio.it (M.C.); 2Laboratory of Medical Genetics, IRCCS-Bambino Gesù Children's Hospital, 00146 Rome, Italy; emanuele.agolini@opbg.net (E.A.); vale17l@gmail.com (V.M.S.); antonio.novelli@opbg.net (A.N.); 3Unit of Bioinformatics, Fondazione IRCCS-Casa Sollievo della Sofferenza, 71043 San Giovanni Rotondo (Foggia), Italy; t.mazza@operapadrepio.it

**Keywords:** Classical-like, Ehlers-Danlos syndrome, haploinsufficiency, tenascin-X, *TNXB*

## Abstract

*TNXB*-related classical-like Ehlers-Danlos syndrome (*TNXB*-clEDS) is an ultrarare type of Ehlers-Danlos syndrome due to biallelic *null* variants in *TNXB*, encoding tenascin-X. Less than 30 individuals have been reported to date, mostly of Dutch origin and showing a phenotype resembling classical Ehlers-Danlos syndrome without atrophic scarring. *TNXB*-clEDS is likely underdiagnosed due to the complex structure of the *TNXB* locus, a fact that complicates diagnostic molecular testing. Here, we report two unrelated Italian women with *TNXB*-clEDS due to compound heterozygosity for *null* alleles in *TNXB*. Both presented soft and hyperextensible skin, generalized joint hypermobility and related musculoskeletal complications, and chronic constipation. In addition, individual 1 showed progressive finger contractures and shortened metatarsals, while individual 2 manifested recurrent subconjunctival hemorrhages and an event of spontaneous rupture of the brachial vein. Molecular testing found the two previously unreported c.8278C > T p.(Gln2760*) and the c.(2358 + 1_2359 − 1)_(2779 + 1_2780 − 1)del variants in Individual 1, and the novel c.1150dupG p.(Glu384Glyfs*57) and the recurrent c.11435_11524+30del variants in Individual 2. mRNA analysis confirmed that the c.(2358 + 1_2359 − 1)_(2779 + 1_2780 − 1)del variant causes a frameshift leading to a predicted truncated protein [p.(Thr787Glyfs*40)]. This study refines the phenotype recently delineated in association with biallelic *null* alleles in *TNXB*, and adds three novel variants to its mutational repertoire. Unusual digital anomalies seem confirmed as possibly peculiar of *TNXB*-clEDS, while vascular fragility could be more than a chance association also in this Ehlers-Danlos syndrome type.

## 1. Introduction

Ehlers-Danlos syndromes (EDS) are a clinically variable and genetically heterogeneous group of hereditary connective tissue disorders mainly featured by abnormal skin texture and repair, tissue and vascular fragility, and joint hypermobility. The 2017 international classification identifies 13 EDS types due to deleterious variants in 19 different genes [1]. More recently, a 14th type of EDS with features overlapping classical type and due to recessive variants in the *AEBP1* gene was added to this nosology [2,3]. Among them, classical, vascular, and hypermobile EDS are the most common, while the others are rarer and their frequency in the general population remains mostly unknown. 

Classical-like EDS due to biallelic variants in *TNXB*, encoding tenascin-X (*TNXB*-clEDS), is a second EDS type resembling classical EDS but distinguished from the latter by recessive pattern of inheritance and lack of atrophic scarring [1]. Tenascin-X is a large extracellular matrix–forming glycoprotein, whose deficiency was firstly involved in the etiology of EDS in 1997, by the description of a novel contiguous gene syndrome combining congenital adrenal hyperplasia (CAH) and EDS, and due to the deletion of the *CYP21A2* (OMIM 201910) and *TNXB* neighbouring genes [4]. Subsequently, it was clear that this rare type of EDS is a recessive disorder caused by homozygous or compound heterozygous *null* alleles in *TNXB* [5]. Only 24 individuals with *TNXB*-clEDS have been described to date and most of them are of Dutch origin [6]. A recent cross-sectional study on 17 individuals suggests possible phenotypic hallmarks of *TNXB*-clEDS and hypothesizes that it is underdiagnosed with EDS-like symptoms outside The Netherlands due to the complex molecular structure of the *TNXB* locus [7]. 

*TNXB* maps on chromosome region 6p23.1 within the human leukocyte antigen histocompatibility complex in a module characterized by highly homologous sequences between functional genes, *CYP21A2* and *TNXB*, and their corresponding pseudogenes *CYP21A1P* and *TNXA*. This genomic structure is prone to non-homologous recombinations. Misalignment events during meiosis result in pathogenic *CYP21A2/CYP21A1P* and *TNXA/TNXB* chimeric genes. To date, three major types of *TNXA/TNXB* chimera have been identified [4,8]. In particular, CAH-X chimera 1 (CH-1) and CAH-X chimera 2 (CH-2) have *TNXB* exons 35–44 and 40–44, respectively, replaced with *TNXA* [4,9,10]. CH-1 is characterized by a 120-bp deletion (c.11435_11524 + 30del) due to the substitution of *TNXB* exon 35 by *TNXA* that is causative of tenascin-X haploinsufficiency in CAH-X CH-1; this region is the only well-documented discrepancy between *TNXB* and *TNXA* homologous portion. CH-2 is characterized by the variant c.12174C > G p.(Cys4058Trp) derived from the substitution of *TNXB* exon 40 by *TNXA* with a likely dominant negative effect [10]. The third chimera, termed CAH-X chimera 3 (CH-3), has *TNXB* exons 41–44 substituted by *TNXA* and is characterized by a cluster of three pseudogene variants: c.12218G > A p.(Arg4073His) in exon 41, and c.12514G > A p.(Asp4172Asn), and c.12524G > A p.(Ser4175Asn) in exon 43. This chimera has been reported in one patient, and its clinical significance is still preliminary [8].

Due to such a complex molecular architecture of the genomic region surrounding *TNXB*, Demirdas et al. [7] proposed a multistep molecular diagnostics workflow including: (i) a mixed approach of next-generation sequencing (NGS) for the non-homologous *TNXB* sequence and Sanger sequencing for the *TNXA/TNXB* homology region to exclude *TNXB* point variants and rare *TNXA/TNXB* chimera, (ii) followed by *TNXB* deletion/duplication analysis aimed to investigate the presence of the recurrent c.11435_11524 + 30del resulting from the common *TNXA/TNXB* chimeric fusion and other rarer rearrangements. Following this approach, this group was able to identify 12 different *TNXB* variants associated with *TNXB*-clEDS [7]. 

Here, we describe the first two Italian individuals affected by *TNXB*-clEDS. Molecular testing investigated the full range of possible molecular mechanisms leading to *TNXB* null alleles and found three novel variants. 

## 2. Materials and Methods 

Patients were enrolled for this study after obtaining written informed consent for publishing pictures (individual 2) and clinical data (both individuals). This study was approved by the local ethical committee, and is in accordance with the 1984 Helsinki declaration and subsequent versions. Part of the results of molecular investigations presented in this work was obtained from the routine clinical diagnostic activities of the involved institutions.

### 2.1. Sample Preparation and Next Generation Sequencing Analysis

Genomic DNA was extracted from patients’ and unaffected relatives’ peripheral blood leucocytes by using Bio Robot EZ1 (Qiagen, Hilden, Germany) according to the manufacturer’s instructions. The DNA was quantified with Qubit spectrophotometer (Thermo Fisher Scientific, Waltham, MA, USA). Probands’ DNA first underwent sequencing with a HaloPlex gene panel (Agilent Technologies, Santa Clara, CA, USA) designed to selectively capture known genes associated with the various forms of EDS (*ADAMTS2, AEBP1, B3GALT6, B4GALT7, CHST14, COL1A1, COL1A2, COL3A1, COL5A1, COL5A2, COL12A1, C1R, C1S, DSE, FKBP14, PLOD1, FLNA, PRDM5, SLC39A13, TNXB* and *ZNF469*) according to the current nosology. For *TNXB*, NGS sequencing is effective for exons 1 to 31 only, which correspond to the non-homologous *TNXB* sequence. Libraries were prepared using the Haloplex target enrichment kit (Agilent Technologies, Santa Clara, CA, USA) following manufacturer’s instructions. Targeted fragments were then sequenced on a MiSeq sequencer (Illumina, San Diego, CA, USA) using a MiSeq Reagent kit V3 300 cycles flow cell. Reads were aligned to the GRCh37/hg19 reference genome by BWA (v.0.7.17). BAM files were sorted by SAMtools (v.1.7) and purged from duplicates using Mark Duplicates from the Picard suite (v.2.9.0). Mapped reads were locally realigned using GATK 3.8. Reads with mapping quality scores lower than 20 or with more than one-half nucleotides with quality scores less than 30 were filtered out. The GATK’s Haplotype Caller tool was used to identify variants. Variants were annotated by ANNOVAR, with information about allelic frequency (1000 Genomes, dbSNP 151, GO-ESP 6500, ExAC, TOPMED, GnomAD, NCI60, COSMIC), reported or computationally estimated pathogenicity (ClinVar, HGMD, LOVD, or SIFT, Polyphen2, LRT, MutationTaster, MutationAssessor, FATHMM, PROVEAN, VEST3, MetaSVM, MetaLR, M-CAP, CADD, DANN, fathmm-MKL, Eigen, GenoCanyon), and amino acids conservation (fitCons, GERP++, phyloP100way, phyloP20way, phastCons100way vertebrate, phastCons20way mammalian, SiPhy 29way). Selected variants were interpreted according to the American College of Medical Genetics and Genomics/Association for Molecular Pathology (ACMGG/AMP) [11]. Specifically, variants without clinical significance at the time of reporting (i.e., benign and likely benign) were excluded by the presence of one or more criteria for benignity. Variants which passed this preliminary selection were selected for further investigation and classified as pathogenic, likely pathogenic or uncertain significance by using the following criteria: (i) null (nonsense, frameshift) variant in a gene previously described as disease-causing by haploinsufficiency or loss-of-function; (ii) missense variant located in a critical and well-established functional domain; (iii) variant affecting canonical splicing sites (i.e., ±1 or ±2 positions); (iv) variant absent in allele frequency population databases; (v) variant reported in allele frequency population databases but with a minor allele frequency significantly lower than the known disease frequency in the general population; (vi) variant predicted as pathogenic/deleterious in ClinVar and/or LOVD; (vii) missense variant predicted as pathogenic/deleterious in most (≥75%) of the selected *in silico* predictors; (viii) variant co-segregating in two or more affected relatives; (ix) the predicted pathogenic effect has been confirmed by an appropriate functional study/studies.

### 2.2. Sanger Sequencing. 

The variants identified by NGS were confirmed by Sanger sequencing. Primer sequences are reported in Table 1. The amplified products were subsequently purified by using ExoSAP-IT PCR Product Cleanup Reagent (Thermo Fisher Scientific, Waltham, MA, USA) and sequenced by using BigDye Terminator v1.1 sequencing kit (Thermo Fisher Scientific, Waltham, MA, USA). The fragments obtained were purified using DyeEx plates (Qiagen, Hilden, Germany) and resolved on ABI Prism 3130 Genetic Analyzer (Thermo Fisher Scientific, Waltham, MA, USA). Sequences were analyzed using the Sequencer software (Gene Codes, Ann Arbor, MI, USA). The identified variant was resequenced in independent experiments.

### 2.3. Analysis of the TNXA/TNXB Homology Region

For the analysis of the *TNXA/TNXB* homology region, the whole genomic sequence of *TNXB,* encompassing exons 32–44 was amplified by employing a Long-PCR reaction using the primers reported in Table 1 and the protocol previously reported [7]. The PCR reaction was performed in a total 50 uL volume reaction containing 5 uL Buffer 1 (10 ×), dNTP (0.625 mM final concentration), primers (10 pmol each), 3U Expand Long Template Enzyme mix (Roche, San Francisco, CA, USA). Cycling conditions were as follows: initial denaturation at 95 °C, 3 min, followed by 30 cycles of 95 °C, 30 s, 62 °C, 30 s, 72 °C, 11 min, final elongation at 72 °C, 7 min. The 4987 bp PCR product was checked on ethidium bromide (EtBr) stained 1% agarose gel and then used as template for seven nested PCR amplifications with the primers listed in Table 1. Reactions were performed in a 25 uL volume reaction containing 2.5 uL Buffer (10 ×), dNTP (0,25 mM final concentration), primers (10 pmol each), 1.25 U AmpliTaq Gold DNA Polymerase (Thermo Fisher Scientific, Waltham, MA, USA). Cycling conditions were the following: initial denaturation at 95 °C, 10 min, followed by 30 cycles of 95 °C, 30 s, 60 °C, 30 s, 72 °C, 30 s, final elongation at 72 °C, 7 min. The overlapping Nested PCR products were then checked on EtBr stained 2% agarose gel, purified with ExoSap-IT kit (Thermo Fisher Scientific, Waltham, MA, USA) and sequenced with a ready reaction kit (BigDye Terminator v1.1 Cycle kit, Thermo Fisher Scientific, Waltham, MA, USA).

### 2.4. Multiplex Ligation-Dependent Probe Amplification (MLPA) and Quantitative PCR (qPCR) Analysis

MLPA was carried out using the commercial kit (SALSA MLPA KIT P155-D2 Ehlers-Danlos syndrome III & IV, MRC Holland, Amsterdam, The Netherlands). The kit includes 17 probes for *TNXB* with amplification products between 130 and 490 nucleotides. This kit also comprises two probes mapping within the region upstream of *TNXB*, located in the *ATF6B* and *BAK1* genes, and two additional probes mapping within *CYP21A2*. Complete probe sequences and the identity of the genes detected by the reference probes is available online. Hybridization, ligation, and PCR amplification were performed according to the manufacturer's instructions. DNAs from three healthy individuals were included as controls. Coffalyser. Net software (MRC Holland, Amsterdam, The Netherlands) was used for data analysis. Detected deletion was confirmed by qPCR. Primers designed for the amplification of DNA fragments by qPCR, including *TNXB* exon 4 to 8 probes, were listed in Table 1. The qPCR was performed using Power SYBR Green PCR Master Mix (Thermo Fisher Scientific, Waltham, MA, USA) on ABI 7900HT real time PCR system (Thermo Fisher Scientific, Waltham, MA, USA). Control DNA fragments were located on different chromosomes. Samples were run in triplicate using standard conditions. 

### 2.5. Chromosome Microarrays Analysis (CMA)

CMA of the individual 1 was performed using the CytoScan™ XON array (Thermo Fisher Scientific, Waltham, MA, USA). This array contains 6.85 million empirically selected probes for whole-genome coverage including 6.5 million copy number probes and 300,000 SNP probes for LOH analysis, duo-trio assessment. The sensitivity of the platform is 95% for the detection of exon-level copy number variations (CNVs). The CytoScan™ XON assay was performed according to the manufacturer’s protocol, starting with 100 ng of patient DNA. Data analysis was performed using the Chromosome Analysis Suite software version 4.2 (Thermo Fisher Scientific, Waltham, MA, USA). A CNV was validated if at least 25 contiguous probes showed an abnormal log2 ratio.

### 2.6. Conservation of the Variant.

Evolutionary conservation of p.Gln2760 residue of tenascin-X was investigated with protein sequence alignment generated by Clustal Omega and compared with data provided by UC Santa Cruz Genome Browser 

### 2.7. Total RNA Analysis. 

Total RNA was extracted using RNeasy Mini Kit (Qiagen, Hilden, Germany), treated with DNase-RNase free (Qiagen, Hilden, Germany), quantified by Nanodrop (Thermo Fisher Scientific, Waltham, MA, USA) and reverse-transcribed using QuantiTect Reverse Transcription Kit (Qiagen, Hilden, Germany) according to the manufacturer’s instructions. PCR amplification and Sanger sequencing were carried out with primers listed in Table 1.

### 2.8. Variant Designation.

Nucleotide variant nomenclature follows the format indicated in the Human Genome Variation Society (HGVS) recommendations. DNA variant numbering system refers to cDNA. GenBank Accession Number NM_019105.6 was used as reference sequences for *TNXB*. Nucleotide numbering uses +1 as the A of the ATG translation initiation codon in the reference sequence, with the initiation codon as codon 1.

## 3. Results

### 3.1. Case Report: Individual 1

This is a 25-year-old woman, first child of healthy and unrelated parents. Her younger brother was described as unaffected. She was born at term, from an uneventful pregnancy and uncomplicated vaginal delivery. Birth parameters were normal. Psychomotor development and education were within normal limits. She recalled congenital contortionism (i.e., positive five-point questionnaire). At 3 years of age, the mother noted trigger fingers of multiple digits in both hands, and this was treated surgically at 16 years of age with complete resolution. She also requested ortodontic therapy for dental crowding and high-arched palate. No visual nor auditory troubles were registered. The patient suffered from constipation since infancy. At 14 years of age, she received the diagnosis of rectal prolapse. She always suffered from easy bruising and tendency to arterial hypotension. More recently, the patient requested rheumatological consultation for proneness to soft-tissue injuries and easy fatigability, although she did never complain of dislocations and fractures. According to the patient’s past clinical history, the rheumatologist requested medical genetics consultation for a suspicion of hereditary connective tissue disorder. At examination, she displayed normal upper limb span/height ratio, bilaterally positive wrist sign, bilaterally negative thumb sign, Beighton score (7/9) with genua recurvata, marked hypermobility of the fingers and toes, flatfeet, brachydactyly (due to shortened metatarsals—brachydactyly type E), of the 2nd and 3rd toes bilaterally, brachydactyly type D of the hands (i.e. shortened and broad distal phalanx of the thumbs), soft, doughy and significantly hyperextensible skin, absence of distrophic scars, palmoplantar hyperlinearity, mild palpebral ptosis and bilateral euryblepharon. Of relevance, no other close family members had brachydactyly. Heart ultrasound showed minimal insufficiency of the mitral valve. Bone densitometry showed reduced bone mass at the femoral head (T-score-1.7). Hand X-rays showed “swan neck” deformity of the interphalangeal distal joint of 1st, 2nd and 3rd left fingers. 

### 3.2. Case Report: Individual 2

This is a unique child of unaffected and unrelated parents. Pregnancy and delivery were uneventful, and psychomotor development and education were normal. She came at our medical genetics outpatient service at 26 years of age due to a long history of polyarticular intense pain (visual analogic scale: 7/10) and (sub)luxations of the shoulders, fingers and temporomandibular joints. Additional musculoskeletal and neurological symptoms included moderate pain (visual analogic scale 4-5/10) of the spine, hands and feet, as well as recurrent myalgias, occipital headache, occasional migraine with aura, post-exertional malaise and chronic fatigue. She also reported easy bruising, an episode of spontaneous rupture of the right brachial vein, multiple episodes of ruptures of cystic lesions of the elbows with leakage of a yellowish and filamenotous substance (presumably, molluscoid pseudotumors), eye dryness, recurrent subconjunctival hemorrhages, gastroesophageal reflux with cardia incontinence, esophageal erosions and severe constipation with coprostasis. At examination, at 26 years of age, the patient showed normal anthropometry, Beighton score 7/9, marked hypermobility of the digits, temporomandibular joints and shoulders, bilateral hallux valgus, genua valga, lumbar hyperlordosis, mild thoracic scoliosis, reduced muscle tone, painful movements, soft, doughy and hyperextensible skin (Figure 1a), normal scar formation, residual elbow scars from recurrent ruptures of (likely) mulluscoid pseudotumors (not visible at the time of examination; Figure 1b), a small subcutaneous spheroid of the pretibial region on the left, multiple cutaneous hematomas, piezogenic papules (Figure 1c), palmoplantar hyperlinearity, bluish sclerae, absence of the lingual frenulum and acrocianosis. Heart ultrasound was normal. Bone densitometry showed mildly reduced bone mass at the femural neck. We were also able to carry out physical examination of individual 2’s parents and both had positive Beighton score according to age and sex (i.e., generalized joint hypermobility). The father also showed mildly soft and hyperexensible skin, although both did not report any significantly related complaints. 

### 3.3. Molecular Findings: Individual 1

NGS analysis performed on DNA from individual 1 identified the novel heterozygous c.8278C> T variant located in the exon 24 of *TNXB* (Figure 1d), which is predicted to incorporate a premature stop codon [p.(Gln2760 *)]. No other candidate variants were found in the remaining genes. The c.8278C > T p.(Gln2760 *) variant was not reported in major databases, including dbSNP, ExAC, 1000 Genomes, and GnomAD. This suggests that the variant represents a rare event and was interpreted as likely pathogenic according to the ACMGG/AMP criteria (i.e., a variant absent in population databases and predicted to generate a null allele in a gene previously known as disease-causing with this molecular mechanism). The result was confirmed by direct Sanger sequencing of proband’s DNA (Figure 2a). Protein sequences alignment of the homologous regions including the Gln2760 residue of human *TNXB* was generated by using the Clustal Omega tool and showed that the affected residue was evolutionarily conserved (Figure 2b). The Gln2760 residue is located in the 19th fibronectin domain of tenascin-X (Figure 1d) and, thus, the truncated protein presumably loses the last multiple 19–31 fibronectin domains as well as the fibrinogen C motif. 

As *TNXB*-clEDS is caused by a complete lack of tenascin-X due to biallelic inactivating variants in *TNXB*, in order to detect the potential presence of a second variant in *TNXB*, we simultaneously performed the long PCR/Sanger sequencing analysis of the *TNXA/TNXB* homologous region and MLPA analysis. The long PCR/Sanger sequencing analysis did not reveal any variant in exons 32–44. On the contrary, MLPA analysis detected a *TNXB* intragenic deletion which includes entirely the exon 5 (Figure 2c). To narrow the proximal deletion breakpoints within the region encompassing the exon 5, qPCR analysis was employed on DNA extracted from patient’s, unaffected parents, and control individuals blood. This approach detected a *TNXB* deletion which include both the whole exons 5 and 6 (Figure 2d).

To better molecularly refine the extension of the deletion, we performed a chromosome microarrays analysis using the CytoScan™ XON array. CMA confirmed an interstitial heterozygous microdeletion at chromosome 6p21.33, covered by 62 array probes and spanning 5 Kb, which encompasses the exons 5 and 6 and flanking intronic regions of *TNXB* (Figure 2e). The molecular karyotype of the patient, accordingly with the International System for Human Cytogenomic Nomenclature 2016 was arr [GRCh37] 6p21.33 (32056115_32061375) x1.

Next, we characterized the deletion at the transcriptional level by direct DNA sequencing of in vitro amplified cDNA product generated from total RNA extracted from patients’ peripheral blood leucocytes (Figure 2f). We showed that the variant c.(2358 + 1_2359 − 1)_(2779 + 1_2780 − 1)del generates a frameshift with the insertion of a premature stop codon in exon 7 [p.(Thr787Glyfs*40)] (Figure 1d). In light of its absence in population databases, the predicted generation of a null allele and the subsequent demonstration by mRNA study, this variant was interpreted as pathogenic according to the ACMGG/AMP rules. Segregation analysis in both unaffected parents was performed by Sanger sequencing and MLPA/qPCR analysis. We detected the c.8278C > T and (2358 + 1_2359 − 1)_(2779 + 1_2780 − 1)del variants in the proband’s father and mother, respectively (Figure 2a,d,f). Both *TNXB* c.8278C > T p.(Gln2760 *) and (2358 + 1_2359 − 1)_(2779 + 1_2780 − 1)del p.(Thr787Glyfs*40) variants have been submitted to the LOVD (Leiden Open Variation Database, https://databases.lovd.nl/shared/variants/0000598484, individual ID # 00266303 https://databases.lovd.nl/shared/variants/0000598485, individual ID # 00266303, respectively). 

### 3.4. Molecular Findings: Individual 2

NGS platform targeted for EDS genes revealed that individual 2 carries out a single base deletion c.1150dupG located in exon 3 of *TNXB* (Figure 1d). This heterozygous variant was predicted to generate a premature stop codon at residue 441 [p.(Glu384Glyfs*57)]. No other candidate variants were found in the remaining genes. The c.1150dupG variant was not reported in major databases. Therefore, the variant was interpreted as likely pathogenic according to the ACMGG/AMP guidelines. This result was confirmed by direct Sanger sequencing of proband’s DNA. The novel variant was also detected in the proband’s mother while it was absent in the father (Figure 2g). MLPA analysis (Figure 2h) detected the recurrent pseudogene-derived 120 bps deletion including the exon 35, previously described by Schalkwijk et al. as the likely result of a common *TNXA/TNXB* fusion gene (CAHX-CH1). This variant was inherited from the healthy carrier father.Both TNXB c.1150dupG p.(Glu384Glyfs*57) and c.11435_11524 + 30del variants have been submitted to the LOVD (https://databases.lovd.nl/shared/variants/0000598486, individual ID # 00266304¸ https://databases.lovd.nl/shared/variants/0000598487, individual ID #00266304, respectively). 

## 4. Discussion

In this work, we described the first two Italian patients with *TNXB*-clEDS, confirming a wider distribution of this rare EDS type in Europe, and the efficacy and reproducibility of the diagnostic approach published by Demirdas et al. [7]. We also identified three novel disease-alleles in *TNXB*, which expand the known mutational spectrum of *TNXB* associated with clEDS (Figure 1d). 

These two adults manifest the previously defined phenotypic spectrum of *TNXB*-clEDS. Scarring was apparently normal in our patients, which is in line with the lack of atrophic/dystrophic scarring as a distinguishing feature, together with recessive inheritance, of *TNXB*-clEDS from classical EDS. Intriguingly, individual 2 reported a history of recurrent ruptures of elbow cystic lesions resembling molluscoid pseudotumors, which are additional cutaneous features considered highly suggestive for classical EDS. This expands the cutaneous similarities between *TNXB*-clEDS and classical EDS; a fact that complicates the differential diagnosis of these disorders on clinical groups and reinforces the opportunity to consider *TNXB* molecular testing in all individuals with a clinical diagnosis of classical EDS resulted negative to *COL5A1*, *COL5A2,* and *COL1A1* (recurrent variants) analysis. Individual 2 also testifies for a possible vascular involvement in *TNXB*-clEDS. In fact, this patient reported recurrent subconjunctival hemorrhages, a feature previously annotated in multiples subjects by Demirdas et al. [7], as well as spontaneous rupture of the brachial vein. The latter is an apparently novel feature of *TNXB*-clEDS and could indicate, if confirmed by other observations, a more severe vascular involvement in this condition.

Demirdas et al. [7] pointed out a peculiar appendicular phenotype of *TNXB*-clEDS featured by foot brachydactyly and small joint (apparently acquired) contractures. Our individual 1 supports this hypothesis, as she showed shortened metatarsals and a history of multiple acquired finger contractures with residual swan neck deformities of the left fingers. Furthermore, constipation and evacuation troubles represented major complaints in both individuals. While these features are not rare within the EDS community of syndromes and are highly represented in adults with hypermobile EDS and hypermobility spectrum disorders [12], this observation in *TNXB*-clEDS confirms the opportunity to better investigate the long-range manifestations of these disorders in order to improve quality of life of individuals with EDS. 

To date, a total of 15 different *TNXB* deleterious variants, including the three novel reported in this paper were described. These variants are frameshift (7/15), stop codon (4/15), or splicing (2/15) and lead to the insertion of a premature stop codon with a presumed loss of expression of the protein. Two out of these 15 variants (2/15) are missense which have detrimental effects on the proper protein folding and stability [10,13]. Among the 15 variants, 11 are identified in single patients/families. A 2 bp deletion (c.3290_3291del), a 30Kb deletion generating a *TNXB/TNXA* fusion gene, and a pseudogene-derived missense variant (c.12174C > G) were found in more than one patient (Figure 1d). Nevertheless, the current nosology of EDS and related disorders clearly states that only “*null* alleles” in *TNXB* can be considered causative of *TNXB*-clEDS [1,6]. Therefore, missense *TNXB* variants should be considered supportive of the diagnosis in a clinical setting only if they appear convincing for haploinsufficiency. 

In this study, individual 1 carries two novel *TNXB* variants, c.8278C>T located in the exon 24 and (2358 + 1_2359 − 1)_(2779 + 1_2780 − 1)del which results in the non in frame deletion of whole exons 5 and 6. Both variants are predicted to generate a premature stop codon. Individual 2 is a compound heterozygote for c.1150dupG and c.11435_11524 + 30del variants. The c.1150dupG is a novel variant located in exon 3 of *TNXB* and is predicted to create a premature stop codon. The c.11435_11524 + 30del variant, which abolishes part of exon 35 and intron 35 of *TNXB,* has been previously described and *TNXA/TNXB* chimeric recombination type 1 [5,7,10]. This deleted region represents the only large *TNXB*-specific sequence in the *TNXA*-homolog region of *TNXB*. The *TNXA*-derived variation is a molecular event which often takes place between a functional gene and a pseudogene. Although this *TNXA/TNXB* fusion gene has been previously characterized, its molecular effect is not yet known. However, tenascin-X serum measurement in affected individuals by previous studies indicate that this variant likely results in a *null* allele [5,9]. In this work, we were not able to carry out a serum dosage of tenascin-X in our patients. Nevertheless, we are confident that the molecular features of the identified variants are convincing for the generation of a not functional allele. 

Due to the complex nature of the genomic region spanning around *TNXB*, the underdiagnosis of *TNXB*-clEDS in the routine diagnostic activities of most laboratories is a likely scenario. In fact, molecular testing of *TNXB* is challenging due to the presence of the pseudogene *TNXA,* which is more than 97% identical to *TNXB* at its 3′ end (exons 32–44). With the only exception of exon 35, which partially shows a *TNXB* specific sequence (see above), exon and intron sequences in this region are (nearly) identical in both *TNXB* and *TNXA*. Our experience confirms the need of a multistep and multi-technique approach (comprising NGS for the non-homologous region, Sanger sequencing with a long-PCR and nested-PCR system for the *TNXA/TNXB* homologous segment, and quantitative analysis for intragenic and intergenic rearrangement) for an efficient analysis of the entire *TNXB* coding region, with only slight modifications from the methodology proposed by Demirdas et al. [7] Tenascin-X serum concentration measurement in patients with suspected *TNXB*-clEDS is an alternative in the absence of effective molecular diagnostic facilities. 

In summary, we reported two additional individuals with *TNXB*-clEDS. Our findings support the previously defined phenotype, which shows similarities with classical EDS but also include some possible distinguishing features and potentially underreported, clinically relevant manifestations. We also expanded the mutational spectrum of *TNXB* and highlighted the need of a high level of specialty for an efficacious *TNXB* molecular screening in a clinical setting.

## Figures and Tables

**Figure 1 genes-10-00967-f001:**
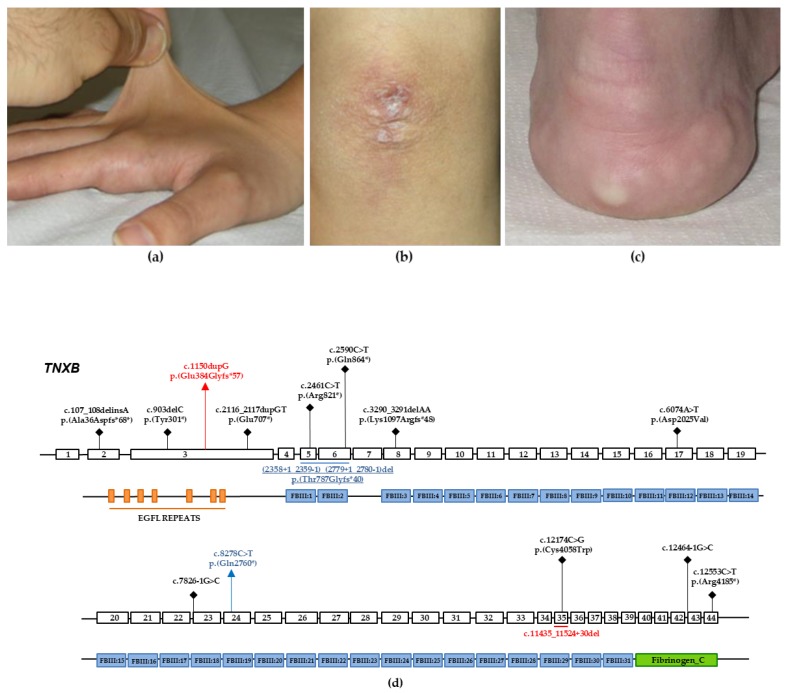
Selected clinical features of individual 2, and schematic diagram showing the genomic structure of *TNXB* and the secondary structure of tenascin-X. (**a**) Markedly hyperextensible skin of the dorsum of the hands. (**b**) Residual scar from recurrent molluscoid pseudotumors of the elbow. (**c**) Piezogenic papules of the heel. (**d**) Coding regions are highlighted with white boxes and introns with black horizontal lines. Tenascin-X is characterized structurally from N-terminus to C-terminus by: (i) an N-terminus with a series of repeats that resemble epidermal growth factor (EGFL repeats); (ii) a stretch of fibronectin type III repeats (FBIII1-31); and (iii) a large C-terminal domain structurally related to fibrinogen (Fibrinogen C). Previously identified variants associated with *TNXB*-clEDS are represented in black. Variants identified in the individuals 1 and 2 are represented in blue and red, respectively.

**Figure 2 genes-10-00967-f002:**
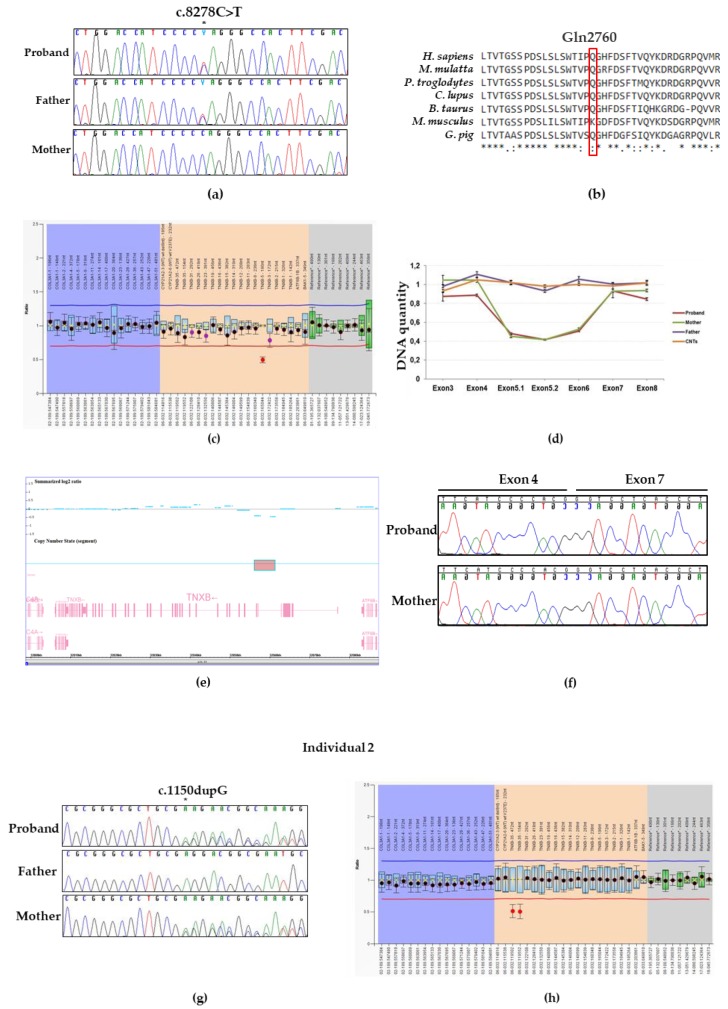
Molecular findings of individuals 1 and 2. (**a**) Electropherograms showing DNA sequencing analysis of PCR product amplified with primers targeting exon 8 of *TNXB.* Nucleotide sequences are provided. The position of the identified variant is indicated with an asterisk. (**b**) Protein sequence alignment of *TNXB* generated by Clustal Omega showed that the affected Gln 2760 residue of tenascin-X is evolutionary conserved. (**c**) Bar chart generated by Coffalyser.net- MLPA analysis software. MLPA was performed on DNA from the individual 1, her unaffected parents and controls (data not showed). A probe ratio of 1 indicates a normal DNA copy number; a probe ratio of 0.5 indicates a heterozygous deletion. MLPA analysis reveals a deletion of exon 5 of *TNXB* in individual 1. (**d**) Profiles of qPCR assay performed to map the deletion breakpoints within the region encompassing the exons 3 to 8 of *TNXB*. Relative DNA quantity of each exon was determined for the patient (red), her asymptomatic mother and father, (green and purple, respectively), and a pool of DNA controls (CNTs, orange). (**e**) Results of chromosomal microarray analysis in the Individual 1. Intensity data (Summarized log 2 ratio value) of each probe is drawn along chromosome 6 from 32,000 to 32,080 kb (USCS Genome Browser build February 2009, hg19). The red box indicates the interstitial microdeletion (62 probes with decreased signal) identified, encompassing the exons 5 and 6 of the TNXB gene (lower panel). (**f**) Electropherograms showing cDNA Sanger sequencing of a transcript region of *TNXB* amplified with primers targeting exon 3 to 8 of Individual 1 and her mother. (**g**) Sanger sequence of a PCR product amplified with primers targeting exon 3 of *TNXB* of individual 2 and her unaffected parents. The position of the identified variant is indicated with an asterisk. (**h**) Bar chart generated by Coffalyser.ne-MLPA analysis performed on DNA from the individual 2, her unaffected father and controls (data not showed). MLPA analysis reveals the common partial deletion of exon 35 of *TNXB* in individual 2.

**Table 1 genes-10-00967-t001:** Primers used in this study.

Primers	Sequence	Tm (°C)	Size (bp)
TNXB-LongPCR-F	GTCTCTGCCCTGGGAATGA	60	4900
TNXB-LongPCR-R	TGTAAACACAGTGCTGCGA		
TNXB frag1 For	GGCCAAGCCTGGAAGATAAA	60	662
TNXB frag1 Rev	GATTGGAGACAGAAGCACAC		
TNXB frag2 For	CCAGGGAGAGAGGATGGAT	60	671
TNXB frag2 Rev	GTCCCCAGGAATGGAAGT		
TNXB frag3 For	GACCTAGTGCCTCAGCCA	60	733
TNXB frag3 Rev	GGCTCTCTCTACTCCGTG		
TNXB frag4 For	ATGGGTGGGAGTTGAGAG	60	727
TNXB frag4 Rev	TGGAAGCTGAGCAGGTAG		
TNXB frag5 For	TCTCCTCTTCCTGCTTTCCC	60	643
TNXB frag5 Rev	CCCCATCAGTCTCCATGTC		
TNXB frag6 For	CAGGACCAGCACCATCTT	60	741
TNXB frag6 Rev	TTGAGGTTGGCGTAGTGG		
TNXB frag7 For	GCTGTCTCCTACCGAGGG	60	621
TNXB frag7 Rev	GCAGAGAAGGCTTCCTCC		
TNXB_EX3Fw	GGTTGCAGTCAGAGGGGGCG	58	300
TNXB_EX3Rv	GCCCGCGACCTCTACAGTCG		
TNXB_EX35Fw	GGGAGCCTCAGAGTGTGC	58	480
TNXB_EX35Rv	TCAGCCCCTGGAGTTTCTG		

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
