# Peer review of "Novel TNXB Variants in Two Italian Patients with Classical-Like Ehlers-Danlos Syndrome"

_genes, 2019, doi:10.3390/genes10120967_

Round 1

Reviewer 1 Report

This is a well worked up report of 2 unrelated cases of a rare type of EDS. It increases the overall knowledge about this cl-EDS with identification of novel mutations and adds to the phenotype.

Minor comments:

'Hyperextensible' skin is a better and more universally recognised term and should be used throughout instead of 'hyperelastic'.

p12 discussion line 389. The authors should perhaps provide more clarity as to why TNX serum measurements are 'technically more elusive' as this is an important potential alternative way to confirm the diagnosis

Author Response

Response to Reviewer 1 Comments

We truly appreciate all the constructive comments and suggestions from this reviewer. We have significantly revised the sections of our manuscript. The following are our point-to-point responses to the reviewers’ comments.

Point 1: 'Hyperextensible' skin is a better and more universally recognised term and should be used throughout instead of 'hyperelastic'.

Response 1: many thanks for this suggestion. The authors modified the text accordingly.

Point 2: p12 discussion line 389. The authors should perhaps provide more clarity as to why TNX serum measurements are 'technically more elusive' as this is an important potential alternative way to confirm the diagnosis

Response 2:

The authors apologize for such a rough sentence. Yes, we agree that biochemical testing may substitute molecular testing in its absence, but the former can also support the latter is presence of variants of unknown significance. The corresponding text was modified accordingly.

Reviewer 2 Report

This is a well-written report describing the molecular characterization of two Italian patients suffering from clEDS. In this study, the authors discovered three previously unreported variants in the gene coding for tenascin-X, underscoring the importance of this gene for the diagnosis of patients suspected for EDS-like conditions. While the study is well-presented and straightforward, a few small points remain to be addressed by the authors.

In the methods the authors mention that the variants identified during sequence analysis were annotated and interpreted using the ACMGG/AMP guidelines. Nevertheless, the final classification of the selected TNXB variants is missing from the manuscript. The authors also mention that no relevant candidate variants were found in the other genes investigated, without giving any more details as to the number of other variants that were not selected for further investigation. Can they elaborate on the specific threshold criteria used to select relevant variants?

Minor comments:

Line 57-58: “A recent cross-sectional study on 17 individuals…” with EDS-like symptoms? Line 76: “still” Line 141: elongation for 30 seconds instead of minutes Line 188: “uneventful” Line 230: should probably be individual 2’s parents Line 231: “hyperextensible” 2b: Why is the comparison with the protein sequence from mus musculus missing? Based on available data, the Gln2760 residue and its neighboring sequence is also conserved in the mouse. Line 332: “this is in line” Line 348 – 452: Were constipation and evacuation troubles also observed in the previously identified TNXB-clEDS patients, or was this finding unique to the Italian patients? Line 354: a verb is missing here

Author Response

Response to Reviewer 2 Comments

We are sincerely grateful to the reviewer as she/he did give us the opportunity to check, correct, and revise these aspects of our work. We have adopted all the suggestions in our revised manuscript. The following are our point-to-point responses to the reviewers’ comments.

Point1: In the methods the authors mention that the variants identified during sequence analysis were annotated and interpreted using the ACMGG/AMP guidelines. Nevertheless, the final classification of the selected TNXB variants is missing from the manuscript. The authors also mention that no relevant candidate variants were found in the other genes investigated, without giving any more details as to the number of other variants that were not selected for further investigation. Can they elaborate on the specific threshold criteria used to select relevant variants?

Response 1: Many thanks for this comment. According to the reviewer’s suggestion, the authors inserted in the methods section the criteria used to select clinically relevant variants.

“Specifically, variants without clinical significance at the time of reporting (i.e., benign and likely benign) were excluded by the presence of one or more criteria for benignity. Variants which passed this preliminary selection were selected for further investigation (pathogenic, likely pathogenic or uncertain significance) by using the following criteria: i) null (nonsense, frameshift) variant in a gene previously described as disease-causing by haploinsufficiency or loss-of-function; ii) missense variant located in a critical and well-established functional domain; iii) variant affecting canonical splicing sites (i.e., ±1 or ±2 positions); iv) variant absent in allele frequency population databases; v) variant reported in allele frequency population databases but with a minor allele frequency significantly lower than the known disease frequency in the general population; vi) variant predicted as pathogenic/deleterious in ClinVar and/or LOVD; vii) missense variant predicted as pathogenic/deleterious in most (≥75%) of the selected in silico predictors; viii) variant co-segregating in two or more affected relatives; ix) the predicted pathogenic effect has been confirmed by appropriate functional study/ies.”

Following reviewer’s suggestion, the authors added the clinical interpretation of the three novel variants according to the ACMGG/AMP guidelines.

Point 2: Line 57-58: “A recent cross-sectional study on 17 individuals…” with EDS-like symptoms?

Response 2: the authors added “with EDS-like symptoms”

Point 3: Line 76: “still”

Response 3: we corrected “till” with “still”

Point 4: Line 141: elongation for 30 seconds instead of minutes

Response 4: we changed “minutes” in “second”

Point 5: Line 188: “uneventful”

Response 5: we corrected “uneventful” with “uneventful”

Point 6: Line 230: should probably be individual 2’s parents

Response 6: the authors are sorry for this mistake. The individual 1 was substituted with the “individual 2”

Point 7: Line 231: “hyperextensible”

Response 7: we changed “hyperexensible” with “hyperextensible”

Point 8: Why is the comparison with the protein sequence from mus musculus missing? Based on available data, the Gln2760 residue and its neighboring sequence is also conserved in the mouse.

Response 8: Thank you for this comment. We aligned the TNXB human and mouse protein sequences by using CLUSTAL Omega but the residue Gln2760 did not result to be conserved between the two species (see the output data). However, the region surrounding the residue Gln2760 is conserved. We modified the Figure 2b accordingly.

Point 9: Line 332: “this is in line”

Response 9: we corrected “this is line” with “this is in line”

Point 9: Line 332: “this is in line”

Response 9: we changed “this is line” with “this is in line”

Point 10: Line 348 – 452: Were constipation and evacuation troubles also observed in the previously identified TNXB-clEDS patients, or was this finding unique to the Italian patients?

Response 10: The authors are sorry for such a lack of details. At the best of our knowledge, these findings are novel in TNXB-clEDS. Nevertheless, these symptoms are common in the general population as well as in the hEDS/HSD community. Therefore, we did not give a high relevance to this finding in the original version of the manuscript. However, the evidence that these features were not explicitly reported in the patients’ cohort by Demirdas et al. has been annotated in the revised version of the manuscript.

Point 11: Line 354: a verb is missing here

Response 11: we added the verb “were described”
